# Trends in the Prevalence and Progression of Diabetic Retinopathy Associated with Hyperglycemic Disorders during Pregnancy in Japan

**DOI:** 10.3390/jcm11010165

**Published:** 2021-12-29

**Authors:** Masahiko Sugimoto, Kohei Sampa, Hideyuki Tsukitome, Kumiko Kato, Hisashi Matsubara, Shin Asami, Kaori Sekimoto, Shigehiko Kitano, Shigeo Yoshida, Yoshihiro Takamura, Takao Hirano, Toshinori Murata, Miho Shimizu, Takamasa Kinoshita, Sentaro Kusuhara, Osamu Sawada, Masahito Ohji, Rina Yoshikawa, Kazuhiro Kimura, Hiroto Ishikawa, Fumi Gomi, Hiroto Terasaki, Mineo Kondo, Tomoaki Ikeda

**Affiliations:** 1Department of Ophthalmology, Graduate School of Medicine, Mie University, Tsu 514-8507, Japan; k.sampa0525@gmail.com (K.S.); dyrkn618@yahoo.co.jp (H.T.); cqw14171jp@yahoo.co.jp (K.K.); hmatsu@clin.medic.mie-u.ac.jp (H.M.); mineo@clin.medic.mie-u.ac.jp (M.K.); 2Graduate School of Medicine, Faculty of Medicine, Mie University, Tsu 514-8507, Japan; 318002@m.mie-u.ac.jp; 3Department of Ophthalmology, Diabetes Center, Tokyo Women’s Medical University, Tokyo 162-8666, Japan; sekimoto.dmc@twmu.ac.jp (K.S.); ge2s-ktn@asahi-net.or.jp (S.K.); 4Department of Ophthalmology, Graduate School of Medicine, Kurume University, Kurume 830-0011, Japan; usyosi@gmail.com; 5Department of Ophthalmology, School of Medical Sciences, University of Fukui, Yoshida 910-1193, Japan; ytakamura@hotmail.com; 6Department of Ophthalmology, School of Medicine, Shinshu University, Matsumoto 390-8621, Japan; takaoh@shinshu-u.ac.jp (T.H.); murata@shinshu-u.ac.jp (T.M.); 7Department of Ophthalmology, Sapporo City General Hospital, Sapporo 060-8604, Japan; miho.shimizu@doc.city.sapporo.jpn (M.S.); knst129@gmail.com (T.K.); 8Division of Ophthalmology, Department of Surgery, Graduate School of Medicine, Kobe University, Kobe 650-0017, Japan; kusu@med.kobe-u.ac.jp; 9Department of Ophthalmology, Shiga University of Medical Science, Otsu 520-2192, Japan; eye.sawada@gmail.com (O.S.); ohji@belle.shiga-med.ac.jp (M.O.); 10Department of Ophthalmology, Graduate School of Medicine, Yamaguchi University, Ube 755-8505, Japan; rinayskw@yamaguchi-u.ac.jp (R.Y.); k.kimura@yamaguchi-u.ac.jp (K.K.); 11Department of Ophthalmology, Hyogo College of Medicine, Nishinomiya 663-8501, Japan; ohmyeye@gmail.com (H.I.); fgomi@hyo-med.ac.jp (F.G.); 12Department of Ophthalmology, Graduate School of Medical and Dental Sciences, Kagoshima University, Kagoshima 890-8520, Japan; teracchi16@yahoo.co.jp; 13Department of Obstetrics and Gynecology, Graduate School of Medicine, Mie University, Tsu 514-8507, Japan; t-ikeda@clin.medic.mie-u.ac.jp

**Keywords:** gestational diabetes mellitus, hyperglycemic disorders during pregnancy, overt diabetes mellitus, pre-existing diabetes mellitus

## Abstract

The aim of this study was to determine the prevalence and progression of diabetic retinopathy (DR) with hyperglycemic disorders during pregnancy (HDPs) in Japan between 2013 and 2018 using two cohorts. The patients with HDPs were classified as those with pre-existing DM (pexD), gestational DM (GDM), and overt DM (ODM). Cohort 1 was obtained from the health claims database whose diseases were classified by the International Classification of Diseases-10. Cohort 2 was derived from a retrospective, multicenter analysis of the medical records of 225 patients from 10 ophthalmological institutions. In Cohort 1, there were 5268 patients with an HDP prevalence of 8.4%. Among them, 73 of 1139 patients had pexD (6.4%) and 61 of 4129 patients with GDM (1.5%) had DR; the overall prevalence of DR was 2.5%. In Cohort 2, 36 of 225 patients (16.0%) had DR, and 149 patients were followed at the early and late stages of pregnancy. Moreover, 10 of the 102 patients with pexD (9.8%) and two of five patients with ODM (40.0%) had a progression of DR. In conclusion, the prevalence and progression of DR in patients with pexD is lower than previously reported. More attention should be given to pexD and ODM.

## 1. Introduction

The number of patients with diabetes mellitus (DM) in the world is increasing; the International Diabetes Federation estimated that 463 million people were living with DM in 2019. It has also been estimated that 20.4 million women who gave birth in 2019 had hyperglycemic disorder during their pregnancy (HDP) (https://www.idf.org/e-library/epidemiology-research/diabetes-atlas/159-idf-diabetes-atlas-ninth-edition-2019.html, accessed on 15 October 2021). Various complications can develop in these patients with HDP. In general, specific risks of uncontrolled diabetes in pregnancy include spontaneous abortion, fetal anomalies, preeclampsia, fetal demise, macrosomia, neonatal hypoglycemia, and neonatal hyperbilirubinemia [1]. Therefore, strict glycemic control is recommended during pregnancy to prevent these complications [2]. In the past, the prognosis of patients with microvascular complications was poor, and termination of the pregnancy was recommended. However, due to improvements in glycemic control, the perinatal mortality of infants has decreased from 10.8% (in 1971–1975) to 1.1% (in 1986–1990) [3].

HDP is classified as pre-existing DM (pexD) and two types of glucose intolerance, gestational diabetes mellitus (GDM) and overt diabetes mellitus (ODM) [4]. Among the patients with HDP, 8.5% were pexD, 83.6% were GDM, and 7.9% were ODM (https://www.idf.org/e-library/epidemiology-research/diabetes-atlas/159-idf-diabetes-atlas-ninth-edition-2019.html, accessed on 15 October 2021). The pexD is present in patients who had been diagnosed as DM, including type 1 and type 2 DM, before the pregnancy. Particularly, the number of women with type 2 DM in pregnancy exceeds that of women with type 1 DM [5]. One multi-institutional retrospective study reported a comparison of pregnancy outcomes between women with type 1 DM and type 2 DM in Japan during 2003−2009. There were no differences in the rates of perinatal mortality and congenital malformation between pregnant women with type 1 DM and type 2 DM. However, women with type 2 DM displayed a higher risk of primary caesarean section [6].

GDM is considered as any degree of glucose intolerance after the first detection of pregnancy and is defined as “any degree of glucose intolerance with onset or first recognition during pregnancy”. The initial criteria for its diagnosis were established more than 40 years ago, although have been modified since. These criteria were chosen to identify women at high risk of developing of DM after pregnancy or were derived from the criteria used for nonpregnant individuals [4]. The biochemical pathways related to the development of GDM are not clear. However, maternal insulin resistance, low-grade inflammation, and endothelial cell dysfunction are central features of GDM, with various candidate regulators such as nuclear factor-κB, peroxisome proliferator-activated receptors, sirtuins, AMP-activated protein kinase, glycogen synthase kinase 3, PI3K/mTOR, inflammasome, and endoplasmic reticulum stress [7]. However, there is a possibility that unrecognized glucose intolerance antedated the pregnancy is not excluded. There is a need to identify these women and address perinatal risks. Thus, the International Association of Diabetes and Pregnancy Study Groups (IADPSG) proposed a definition for ODM: “pregnant women who met the criteria for DM in the nonpregnant state but were not previously diagnosed with DM”. ODM has many differences compared with GDM: (1) increased risk of congenital anomalies in offspring, (2) risk of diabetic complications requiring treatment during pregnancy, (3) a need for rapid treatment and close follow-up during pregnancy to ensure the prompt restoration of normal glycemia, and (4) a need to ensure confirmation and appropriate treatment of DM after pregnancy [4]. The Japan Diabetes and Pregnancy Study Group performed a multi-institutional retrospective study and reported that people with ODM exhibited significantly higher HbA1c levels and a higher incidence of maternal complications compared with people with GDM [8].

Pregnancy is a risk factor for the progression of diabetic retinopathy (DR). The odds ratio (OR) for DR progression in pregnant women is higher compared with nonpregnant women [9]. Morrison et al. reviewed the prevalence and progression of DR with pexD [10]. The prevalence of DR in early pregnancy was 14% in type 2 DM (T2DM) [11] and up to 63% in type 1 DM (T1DM) [12]. However, these studies were reported in European or other Western countries, and do not reflect recent trends in Japan. There has been no large-scale analysis on the prevalence and progression of DR for pregnant women in Japan. Thus, the purpose of this study was to perform a retrospective, large-scale analysis of Japanese women with HDP to determine the prevalence and progression of DR.

## 2. Patients and Methods

This study consisted of two Japanese cohorts. The study protocol conformed to the tenets of the Declaration of Helsinki for research involving human subjects. The protocol was approved by the Institutional Review Board of Mie University Graduate School of Medicine (H2019-066 and H2020-076), and it conformed to the principles of Good Clinical Practice and the Helsinki Guidelines. This prospective study was registered online at http://www.umin.ac.jp (UMIN ID 000037670 and UMIN ID 000039377).

### 2.1. Study Design and Source of Data for Cohort 1

The study design of Cohort 1 was a retrospective observational analysis of the data of anonymized individuals obtained from the health claims database of the Japan Medical Data Center (JMDC, Tokyo, Japan). The data were registered between 2013 and 2018 and contained medical claims provided by multiple health insurance services. The database included 7,317,946 individuals consisting of 3,894,276 men and 3,423,670 women. They belonged to a health insurance provider for company employees and their dependents in Japan. This nationwide claim database included the demographics and diagnosis codes, recorded as standardized local codes of diseases defined by the Medical Information System Development Center and mapped to the International Classification of Diseases (ICD)-10 codes. The data contained in this database were anonymized; therefore, the requirement of individual informed consent was waived. Details of the study protocol have been provided elsewhere [13].

### 2.2. Study Participants of Cohort 1

Cohort 1 consisted of pregnant women (age ≥ 18 years) diagnosed as GDM, T1DM, T2DM, or DM (ICD-10 codes E10, E11, E14; O240, O241, O244; and O249, respectively) between 1 January 2013, and 31 December 2018. All were registered in the JMDC more than 1 year before the pregnancy and observed from the early stage to 1 month after delivery. Pregnancy was determined and accepted if a single pregnancy was confirmed in this period. A woman with pexD was defined as presenting with DM before the diagnosis of pregnancy. GDM was defined as an absence of DM at least 1 year before the index date, and the existence of DM was determined after the diagnosis of pregnancy and before the delivery. There is no ICD-10 definition of ODM; therefore, a part of ODM was included in the GDM or pexD for this cohort.

### 2.3. Definitions of Outcomes

The first occurrence of DR was confirmed by ICD-10 codes E103, E113, E143, H356, H360, H431, or H450. For the treatment of DR, laser retinal photocoagulation was identified by the healthcare insurance claim code K276. Vitrectomy and retinal surgeries were determined by healthcare insurance claim codes K278, K279, K280, K280-2, K281, and K284. Treatment with antivascular endothelial growth factor (anti-VEGF) agents was confirmed by prescriptions of ranibizumab or aflibercept (ATC codes, S01P). Steroid treatments were confirmed by prescriptions of triamcinolone acetonide (ATC codes, S01B or H02A1).

Systemic complications during pregnancy were defined as preeclampsia, hypertension during pregnancy, and renal failure. Complications during pregnancy were determined by ICD-10 codes I110, I119, I120, I129, I139, I150, I151, I152, I158, I159, O100, O101, O102, O103, O104, O120, O121, O140, O141, O142, O149, O150, O151, O152, and O159. Processes of obstetric delivery other than normal delivery were defined as caesarean section and induced labor. Caesarean section was determined by ICD-10 codes O820, O821, and O829. Induced labor was defined as a delivery with use of agents, with an ATC code of G02A.

### 2.4. Study Design and Source of Data for Cohort 2

Data for the individuals in Cohort 2 were the longitudinal medical records for a demographically diverse patient population obtained from 20 retina specialists at 10 ophthalmological institutions in Japan. This cohort was identified by the existence of DR or the progression of DR during pregnancy to determine the risk of cumulative incidences of DR during the follow-up period. This retrospective observational study included all eligible patients who had been diagnosed with HDP between January 2013 and December 2018. They were followed during one pregnancy from early stage to 1 month after the delivery.

### 2.5. Subjects for Cohort 2

For this registry, patients were recruited from the Department of Ophthalmology and Department of Obstetrics Gynecology during the gestational period. They were classified as pexD, GDM, and ODM using the Japan Diabetes Society criteria for HDP based on the international criteria stated by the IADPSG [4]. The gestational week was estimated based on the findings from the early ultrasound scanning images. The patients were divided into: (1) early stage, <27 weeks of pregnancy and in the first and second trimesters; and (2) late stage, ≥27 weeks of pregnancy, in the third trimester. Women with multifetal gestations, previous diagnosis of GDM or an active chronic systemic disease other than chronic hypertension or renal disfunction, and those with the second pregnancies in the same year were excluded. The data obtained from the medical records included the age, type of DM with pexD, duration of the DM with pexD, DR stage, blood pressure, serum creatinine levels, and the gestational age. The systolic blood pressure (SBP) and diastolic blood pressure (DBP) were measured in the upper arm with a sphygmomanometer. The hemoglobin A1c (HbA1c) levels were measured by column chromatography with the nationally standardized glyco-hemoglobin program. The degree of nephropathy was determined from the serum creatinine levels (Cr).

### 2.6. Ophthalmic Examinations

The patients were examined in the Department of Ophthalmology of each hospital during the gestational period. All patients underwent comprehensive ophthalmologic examinations including slit lamp biomicroscopy and fundus examination with dilated pupils using 1% tropicamide and 2.5% phenylephrine hydrochloride. The fundus was examined by trained retinal specialists using indirect ophthalmoscopy, and the severity of DR was classified into five categories: no DR (NDR), mild nonproliferative DR (mild NPDR), moderate NPDR, severe NPDR, and proliferative DR (PDR), according to the International Clinical Diabetic Retinopathy Disease Severity Scale [14]. For eyes with bilateral DR, we used the more severe side. The examinations were generally performed at the early and late stages of pregnancy. A progression of DR was defined as an advancement of one stage of DR in at least one eye.

### 2.7. Statistical Analyses

All values are presented as the means ± standard deviations. The prevalence of DR was estimated by dividing the number of patients with DR by the number of each type of HDP. The OR and 95% confidence intervals (95% CIs), chi-squared tests, and Cramer’s coefficient of contingency were calculated to estimate the prevalence of DR for each type of HDP. Multiple logistic regression analysis and analysis of covariance (ANCOVA) were performed to adjust for potential confounding factors including the age, HbA1c, gestational weeks, SBP, DBP, Cr, and insulin use. Tukey-type multiple comparisons was used to compare the adjusted means. The threshold for statistical significance was set at *p* < 0.05.

## 3. Results

### 3.1. Representative Data for Cohort 1 from Health Claims of Japan Medical Data Center (JMDC) Database

There were 62,897 women identified as pregnant during this period. Among them, 6921 women developed HDP, representing a prevalence of 8.4%. It was difficult to identify the type of HDP in 1653 patients with diabetes; therefore, 5268 women whose DMDP could be classified were studied. Among them, 1139 had pexD and 4129 had GDM. The demographics of these patients are shown in Appendix A. The average age of the whole patient cohort was 33.8 ± 4.5 years, that of those with pexD was 34.2 ± 4.6 years, and that of those with GDM was 33.7 ± 4.5 years. Seventy-three patients with pexD had DR, with an average age of 34.4 ± 4.6 years, and sixty-one patients with GDM had DR, with an average age of 35.4 ± 4.5 years.

Table 1 shows the prevalence of DR from the Cohort 1 data. The prevalence of DR for all the HDP patients was 2.5%. The prevalence of DR in the pexD group was 6.4% and that in the GDM group was 1.5%. The prevalence of DR was significant higher in the pexD group than in the GDM group, with an OR of 4.6 (95% CI 3.2–6.5, *p* = 1.44 × 10^−15^; Table 1). None of the patients with pexD or GDM received any postpartum ophthalmic treatments. We also compared the type of maternal complications in this cohort relative to the presence of DR (Table 2). The incidence of maternal complications associated with DR was significantly high for pexD compared with GDM: hypertension (OR of 6.0 (95% CI 2.5–15.0), *p* = 9.78 × 10^−5^), renal failure (OR of 22.5 (95% CI 2.8–183.7), *p* = 0.003), caesarean section (OR of 9.0 (95% CI 4.4–18.1), *p* = 1.19 × 10^−9^), and induced labor (OR of 6.5 (95% CI 3.8–11.0), *p* = 6.05 × 10^−12^). There were no significant differences in the prevalence of ophthalmic complications associated with the presence of DR between the HDP types: cataract (*p* = 0.26), glaucoma (*p* = 0.64), and others (*p* = 0.23).

### 3.2. Representative Data from Patients of Cohort 2

The demographics and DR stage of Cohort 2 obtained from our patient registry are presented in Appendix A. There were 225 patients identified with HDP in all groups: 119 patients with pexD (67 patients with T1DM and 52 patients with T2DM), 96 patients with GDM, and 10 patients with ODM. The prevalence of DR across all the groups was 16.0% (36/225). We compared the stage of DR for each group using multiple logistic regression analyses (Table 3). Adjustments were made for age, HbA1c, gestational weeks, SBP, DBP, Cr, and insulin use. Our results showed no significant association between any of these factors with the stage of DR (*p* = 0.40, ANCOVA). There were 31 patients presenting a sign of DR with pexD (26.1%), 2 patients presenting with GDM (2.1%), and 3 patients presenting with ODM (30.0%). The prevalence of DR was significantly higher in the pexD group than in the GDM/ODM groups (*p* = 0.03, chi-squared test). The stage of DR was significantly more severe in the pexD group than in the GDM group (*p* = 0.03, Tukey-type multiple comparison).

### 3.3. Clinical Characteristics of Patients Who Were Evaluated during Early and Late Stages from the Multicenter Database

There were 149 HDP patients in Cohort 2 who could be evaluated at both the early and late stages of pregnancy. We compared DR progression between the groups (Table 4). There were 102 patients with pexD (61 patients with T1DM and 41 patients with T2DM), 42 with GDM, and 5 with ODM. The clinical characteristics of these patients are presented in Table 4. In these groups, no significant changes in HbA1c, SBP, DBP, or Cr were found at the two stages (paired *t*-tests). There were 12 (8.1%) patients that had a progression of DR between the early and late stages; 10 with pexD and 2 with ODM. Among them, 3 cases presenting with pexD progressed from moderate NPDR to severe NPDR and were treated with pan-retinal photocoagulation. No DR progression was observed with GDM.

## 4. Discussion

Our analyses of the two different cohorts enabled the examination of DR development and progression in pregnant women with HDP in Japan. Although the prevalence of DR in patients with HDP was not high in Japan, we still need to give attention to patients with DR with pexD and ODM because the possibility of DR progression. The advantage of Cohort 1 was the huge amount of data from the health claim system. However, there were disadvantages with this cohort. First, our classifications of the types of disease were not strict. In this cohort, there were 1653 unknown patients with a diagnosis of “diabetes during pregnancy”. There is no ICD-10 definition for ODM; therefore, we could not separate ODM from GDM for these unknown patients. This may have affected the true prevalence of DR. Second, the data of Cohort 1 lacked detailed clinical information. In addition, the prevalence of pregnant complications (i.e., hypertension and caesarean section) seems to be low compared to previous studies. To overcome these disadvantages, we established Cohort 2, which comprised data obtained from the medical records of multiple institutions. Although the number of patients was smaller for Cohort 2 (5268 vs. 225), this cohort provided detailed information, including the stage of DR, which was lacking in Cohort 1. We believe that these two cohorts complementarily represented the nationwide status of DR in women with HDP in Japan.

### 4.1. GDM Presents Lower Risk for DR Prevalence and Its Progression

Similar to our result for Cohort 1, the general complications with GDM patients have been reported to be fewer than those of patients with pexD [15]. Earlier studies also found that patients with GDM do not require fundus examinations during pregnancy because the risk for developing DR is very low [16]. *Preferred Practice Patterns*, published by the American Academy of Ophthalmology, stated that women who develop GDM do not have an increased risk for DR during pregnancy and do not require an eye examination (https://www.aao.org/preferred-practice-pattern/diabetic-retinopathy-ppp, accessed on 15 October 2021). In fact, the prevalence of DR in the group of women with GDM was very low in both of our cohorts (1.5% for Cohort I and 2.1% for Cohort 2). The definition of GDM for a Japanese population was established by the Japanese Society of Diabetes and Pregnancy in 2015 based on 2010 International Association of Diabetes and Pregnancy Study Groups criteria. According to this definition, patients with DR during pregnancy are newly categorized as pexD or ODM and separated from GDM. This new definition was established in 2015; therefore, the study period in our analysis may have included two different definitions. In a future study, we need to collect longitudinal data from after 2015 and examine the prevalence of DR under new definitions. However, GDM has other important complications. Although most GDM patients return to normal glycemic control after delivery, their HbA1c levels have been found to be higher than those in normoglycemic women [17], and GDM women have a greater risk of conversion to T2DM during the follow-up period after delivery [18]. There is controversy about the importance of long-term eye care in patients with GDM. Beharier et al. reported the results of long-term observations of 9888 GDM cases in a single institution and concluded that patients with GDM had significantly higher incidences of ophthalmic morbidity [19]. Appropriate follow-up periods should be established for patients with GDM after pregnancy, in accordance with the guidelines for T2DM, and including fundus examinations at the time of diagnosis of conversion to T2DM. Our results also support this; GDM does not require a conventional DR follow-up schedule in Japan.

### 4.2. Consciousness for DR and Its Progression for pexD Management

All our results presented from two cohorts show different aspects of pexD and ODM compared to GDM. The progression of DR during pregnancy, especially for pexD, depends on various factors. Relph et al. reported a meta-analysis of 56 cohort studies involving 12,819 pregnant women with diabetes: 40 studies from Europe and 9 from North America. DR was associated with any preterm birth and preeclampsia. The risks of onset or worsening of DR were increased in women who were nulliparous, smokers, presented with existing proliferative disease, had more chronic diabetes [20]. Our results also show that hypertension, renal failure, eclampsia, caesarean section, and induced labor are general risk factors for DR. However, ophthalmic complications, including cataracts and glaucoma, were not. Therefore, DR is still an important complication for pexD.

Although the development of sight-threatening retinopathy was rare, DR progression occurred occasionally. Morrison et al. [10] reviewed studies on the progression of DR in patients with pexD. They reported that an average prevalence of DR progression was approximately 10.0–22.0% with pexD. Bourry et al. reported DR progression in France with T1DM using longitudinal retrospective data from 1997 to 2015; its progression rate was 15.9–24.4% [21]. They reported that progression mainly occurred in early or mid-pregnancy, and elevated pre-pregnancy HbA1c levels and longer duration of diabetes were predictors of DR progression. However, because most of the studies were from European or Western countries, these results cannot be compared to our findings (9.8% for pexD in Cohort 2). This difference was especially true because the ethnicity is a risk factor for developing HDP [22]. There are only a few reports on the progression of DR in patients with pexD in Japan. Toda et al. reported on the findings in 93 patients with pexD from a single Japanese institution from 2004 to 2010. They concluded that the progression of DR was 17%, lower than that reported in earlier studies [23]. Our results showed that DR progression with pexD was 9.8% for Cohort 2, although PDR did exist during pregnancy. Advanced ophthalmic care may contribute to preventing DR progression to severe stages before and during pregnancy. Compared to previous reports, the progression of DR in patients with pexD was also low. We concluded that DR progression in the pexD group was lower due to better glycemic controls in Japan. Thus, our results confirm the AAO statement: “different from the 1960s, DR is no longer considered as a reason for contraindication to vaginal birth” [24]. However, because DR and its progression occur in patients with pexD, we still need to be conscious about the management of pexD, although cases of severe progression are rare in Japan.

### 4.3. ODM Requires Different Management in DR Compared with GDM

ODM is new definition, and there remain unclear points for DR management in ODM. Sugiyama et al. retrospectively examined 1615 patients with GDM and ODM and maternal complications and reported that the prevalence of DR was significantly higher in ODM patients than in GDM patients. However, there were no detailed evaluations for the characteristics such as the progression of DR [8]. Our results showed a progression of DR with ODM, which is reasonable because the degree of carbohydrate intolerance is more severe in ODM than GDM. In fact, two patients with GDM exhibited mild NPDR. We could not identify the onset of DM clearly from the medical records for ODM patients, and it was difficult to differentiate GDM from ODM. Therefore, as well as for GDM, there should be increased focus on DR with ODM, the same as for with pexD.

### 4.4. Limitations

This study had some limitations. First, because data from health claims were used for Cohort 1, it was not possible to determine whether glycemic control in each group was appropriate. Second, there was a difference in the prevalence of DR between the two cohorts. This is probably due to differences between the health claim database and medical record database. The prevalence varies depending on the study design, including community-based, hospital-based, nationwide, and local designs. Subjects in the JMDC database were employees and their dependents; therefore, it is possible that a number of patients in other categories, e.g., self-employed or public assistance, were not included. In addition, because the sample size of Cohort 2 was small compared to Cohort 1, it was difficult to compare both datasets. Detailed analyses should be obtained from these different databases.

## 5. Conclusions

Using two different national cohorts in Japan, we found that the prevalence and progression of DR in pregnant women with pexD were lower than reported. Although severe DR progression is rare, we need increased attention on DR management with pexD patients. Additionally, we do not need much attention for GDM patients, because DR prevalence and its progression are rare. However, as with pexD, we should develop care for ODM patients. In conclusion, developments should focus on pexD and ODM.

## Figures and Tables

**Table 1 jcm-11-00165-t001:** The prevalence of DR from the Japanese claim database cohort.

Total			DR (+)		DR (−)		Odds Ratio for DR between the Groups	*p*-Value
	N	Age (Years)	N (%)	Age (Years)	N (%)	Age (Years)		
**pexD**	1139	34.2 ± 4.6	73 (6.4)	34.4 ± 4.6	1066 (93.6)	34.2 ± 4.6		
**GDM**	4129	33.7 ± 4.5	61 (1.5)	35.4 ± 4.5	4068 (98.5)	33.7 ± 4.5		
**Total**	5268	33.8 ± 4.5	134 (2.5)	34.8 ± 4.5	5134 (97.5)	33.8 ± 4.5	4.6 (3.2–6.5) **	1.44 × 10^−15^

DR: diabetic retinopathy, GDM: gestational diabetes mellitus, pexD: pre-existing diabetes mellitus. Values are presented as the mean ± standard deviation. ** *p* < 0.01.

**Table 2 jcm-11-00165-t002:** Complications during pregnancy from Japanese claim database cohort.

	pexD		GDM		Odds Ratio (95% CI)	*p*-Value
	DR (+)	DR (−)	DR (+)	DR (−)		
**<General complications>**						
**Hypertension**	13 (11.7%)	98 (88.3%)	8 (2.1%)	367 (97.9%)	6.0 (2.5–15.0) **	9.78 × 10^−5^
**Renal failure**	9 (19.2)	38 (80.9)	1 (1.1)	95 (99.0)	22.5 (2.8–183.7) **	0.003
**Eclampsia**	0 (0)	0 (0)	0 (0)	13 (100)	Not available	Not available
**Caesarean section**	25 (11.7)	189 (88.3)	12 (1.5)	812 (98.5)	9.0 (4.4–18.1) **	1.19 × 10^−9^
**Induced labor**	32 (9.88)	292 (90.1)	6 (1.7)	1537 (98.3)	6.5 (3.8–11.0) **	6.05 × 10^−12^
**<Ophthalmic complications>**						
**Cataract**	1 (100%)	0(0%)	3 (27.3%)	8 (72.7%)	7.3 (0.2–225.9)	0.26
**Glaucoma**	2 (20.0)	8 (80.0)	4 (13.8)	25 (86.2)	1.6 (0.2–10.2)	0.64
**Other**	10 (12.8)	68 (87.2)	28 (8.4)	304 (91.6)	1.6 (0.7–3.4)	0.23

DR: diabetic retinopathy, GDM: gestational diabetes mellitus, pexD: pre-existing diabetes mellitus. Values are presented as the mean ± standard deviation. ** *p* < 0.01.

**Table 3 jcm-11-00165-t003:** The distribution of DR stage from the multicenter cohort.

	N	Age (Years)	NDR	Mild	Moderate	Severe	PDR	Total DR
**pexD**	119	33.4 ± 4.9	88	17	6	4	4	31 (26.1%) *
**GDM**	96	34.3 ± 5.3	94	2	0	0	0	2 (2.1%)
**ODM**	10	30.9 ± 4.6	7	2	1	0	0	3 (30.0%)
**Total**	225	33.6 ± 5.1	189	21	7	4	4	36 (16.0%)

DR: diabetic retinopathy, GDM: gestational diabetes mellitus, NDR: no diabetic retinopathy, ODM: overt diabetes mellitus, pexD: pre-existing diabetes mellitus, PDR: proliferative diabetic retinopathy. Analysis of covariance (ANCOVA) with Tukey-type multiple comparison as a post hoc test was performed. * *p* < 0.05.

**Table 4 jcm-11-00165-t004:** DR progression ratio from multicenter cohort.

	N	Age (Years)	Duration (Years)	DR Progression (%)
**pexD**	102	33.5 ± 4.6	12.9 ± 7.9	10 (9.8)
**GDM**	42	34.1 ± 5.4		0 (0)
**ODM**	5	31.6 ± 3.2		2 (40.0)
**Total**	149	33.6 ± 4.8	12.9 ± 7.9	12 (8.1)

DR: diabetic retinopathy, GDM: gestational diabetes mellitus, ODM: overt diabetes mellitus, pexD: pre-existing diabetes mellitus.

## Data Availability

The data that support the findings of this study are available from the corresponding author upon reasonable request.

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
