# Peer review of "Trends in the Prevalence and Progression of Diabetic Retinopathy Associated with Hyperglycemic Disorders during Pregnancy in Japan"

_jcm, 2021, doi:10.3390/jcm11010165_

Round 1
Reviewer 1 Report
The manuscript studied the relation between prevalence and progression of diabetic retinopathy and hyperglycemic disorders including preexisting DM (pexD), gestational DM (GDM), and overt DM (ODM). The manuscript is interesting
My comments
-The authors should indicate if the statistics reported related to Japan or worldwide. For example line 57
-In introduction, please provide brief differentiation between preexisting DM (pexD), gestational DM (GDM), and overt DM (ODM) in context of clinical and Biochemical presentations. Also, the adverse effects of these conditions on both mother and fetus and/or infant should be more highlighted
Author Response
Dec 3, 2021
To the Editor:
I wish to submit an original research article for publication in Journal of Clinical Medicine®, titled “Trends in the prevalence and progression of diabetic retinopathy associate with hyperglycemic disorders during pregnancy in Japan.”
We have made changes according to reviewers’ comments. Modified points are highlighted in red.
Especially, because all Reviewers recommended English language revision, we asked English editing. We rephase the manuscript as they pointed out.
This manuscript has not been published or presented elsewhere in part or in entirety and is not under consideration by another journal. All study participants provided informed consent, and the study design was approved by the appropriate ethics review board. We have read and understood your journal’s policies, and we believe that neither the manuscript nor the study violates any of these. Details about competing interests are reported in this manuscript.
We thank you for your considerations.
Best regards,
Masahiko Sugimoto, M.D., Ph.D.
Department of Ophthalmology, Mie University Graduate School of Medicine
2-174 Edobashi, Tsu 514-8507, Japan
Phone: +81-59-231-5027
Fax: +81-59-231-3036
E-mail: sugmochi@clin.medic.mie-u.ac.jp
Response to Reviewers
To the Reviewer #1
- The authors should indicate if the statistics reported related to Japan or worldwide. For example, line 57. A) We add explanation in the Introduction (L53-, L65-, L73-).
- In introduction, please provide brief differentiation between preexisting DM (pexD), gestational DM (GDM), and overt DM(ODM) in context of clinical and Biochemical presentations. Also, the adverse effects of these conditions on both mother and fetus and/or infant should be more highlighted. A) We add explanation about them (L53-, L65-, L73-).

Reviewer 2 Report
The authors determine the prevalence and progression of diabetic retinopathy (DR) with hyperglycemic disorders during pregnancy (HDP) in Japan between 2013 and 2018 using two cohort and they found that , the prevalence and progression of DR in patients with pexD is lower than previously reported. We need attention with pexD and ODM.
The article is well-written and well design son issues must be addressed previous to continue with the publication process.
1 – Simplify the title to ensure the key message of the paper is present
2 – Explain in detail examinations in point 2.6
3 – Redesign tables in order to present data more clarify
4 – Explain de practical issue that improve daily ophthalmology routine with the results of this study
5 – Updated the old one references previous to 2005
6 – Exclude references out of indexed journal in Journal Citation Reports when possible
7 – Structure the discussion into section to clarify the message of the topic
Author Response
Dec 3, 2021
To the Editor:
I wish to submit an original research article for publication in Journal of Clinical Medicine®, titled “Trends in the prevalence and progression of diabetic retinopathy associate with hyperglycemic disorders during pregnancy in Japan.”
We have made changes according to reviewers’ comments. Modified points are highlighted in red.
Especially, because all Reviewers recommended English language revision, we asked English editing. We rephase the manuscript as they pointed out.
This manuscript has not been published or presented elsewhere in part or in entirety and is not under consideration by another journal. All study participants provided informed consent, and the study design was approved by the appropriate ethics review board. We have read and understood your journal’s policies, and we believe that neither the manuscript nor the study violates any of these. Details about competing interests are reported in this manuscript.
We thank you for your considerations.
Best regards,
Masahiko Sugimoto, M.D., Ph.D.
Department of Ophthalmology, Mie University Graduate School of Medicine
2-174 Edobashi, Tsu 514-8507, Japan
Phone: +81-59-231-5027
Fax: +81-59-231-3036
E-mail: sugmochi@clin.medic.mie-u.ac.jp
Response to Reviewers
To the Reviewer #2
- Simplify the title to ensure the key message of the paper is present. A) We made change for the tittle.
- Explain in detail examinations in point 2.6. A) We add explanation about fundus examination (L189-).
- Redesign tables in order to present data more clarify. A) We redesign tables and add supplemental tables.
- Explain de practical issue that improve daily ophthalmology routine with the results of this study. A) We add explanation about them (L366-, L376-, L400-, L413-, and “5. Conclusions”).
- Updated the old one references previous to 2005. A) We delete old references except Ref 3 and 14 (because they are important though old references). We also add newer references.
- Exclude references out of indexed journal in Journal Citation Reports when possible. A) We deleted such references.
- Structure the discussion into section to clarify the message of the topic. A) We modified discussion into section.

Reviewer 3 Report
Major concerns:
#1 Title is too long, consider shortening it
#2 The sample of cohort 2 is small compared to cohort 1, it is difficult to compare data.
#3 Figure 1 is little worked.
#4 Tables should present the data more clearly, work your presentation. The legend of the abbreviations is very close to the data.
#5 The conclusion should focus on the study data. Concluding that more follow-up and attention is needed is too general.
Author Response
Dec 3, 2021
To the Editor:
I wish to submit an original research article for publication in Journal of Clinical Medicine®, titled “Trends in the prevalence and progression of diabetic retinopathy associate with hyperglycemic disorders during pregnancy in Japan.”
We have made changes according to reviewers’ comments. Modified points are highlighted in red.
Especially, because all Reviewers recommended English language revision, we asked English editing. We rephase the manuscript as they pointed out.
This manuscript has not been published or presented elsewhere in part or in entirety and is not under consideration by another journal. All study participants provided informed consent, and the study design was approved by the appropriate ethics review board. We have read and understood your journal’s policies, and we believe that neither the manuscript nor the study violates any of these. Details about competing interests are reported in this manuscript.
We thank you for your considerations.
Best regards,
Masahiko Sugimoto, M.D., Ph.D.
Department of Ophthalmology, Mie University Graduate School of Medicine
2-174 Edobashi, Tsu 514-8507, Japan
Phone: +81-59-231-5027
Fax: +81-59-231-3036
E-mail: sugmochi@clin.medic.mie-u.ac.jp
Response to Reviewers
To the Reviewer #3
- Title is too long, consider shortening it. A) We made change for the tittle.
- The sample of cohort 2 is small compared to cohort 1, it is difficult to compare data. A) We agree with this comment and add explanation (L424-).
- Figure 1 is little worked. A) We deleted Figure 1 and its legend.
- Tables should present the data more clearly, work your presentation. The legend of the abbreviations is very close to the data. A) We redesign tables and add supplemental tables.
- The conclusion should focus on the study data. Concluding that more follow-up and attention is needed is too general. A) We changed conclusions (L430-).

Round 2
Reviewer 3 Report
I agree with the corrections that have been made with my suggestions and those of other reviewers.